# Thermocatalytic hydrogen peroxide generation and environmental disinfection by Bi$_2$Te$_3$ nanoplates

Yu-Jiung Lin[1,9], Imran Khan [2,9], Subhajit Saha [1], Chih-Cheng Wu[1,3,4,5], Snigdha Roy Barman[1], Fu-Cheng Kao[1,6] & Zong-Hong Lin [1,7,8 ✉]

The highly reactive nature of reactive oxygen species (ROS) is the basis for widespread use in environmental and health-related fields. Conventionally, there are only two kinds of catalysts used for ROS generation: photocatalysts and piezocatalysts. However, their usage has been limited due to various environmental and physical factors. To address this problem, herein, we report thermoelectric materials, such as Bi$_2$Te$_3$, Sb$_2$Te$_3$, and PbTe, as thermocatalysts which can produce hydrogen peroxide (H$_2$O$_2$) under a small surrounding temperature difference. Being the most prevalent environmental factors in daily life, temperature and related thermal effects have tremendous potential for practical applications. To increase the practicality in everyday life, bismuth telluride nanoplates (Bi$_2$Te$_3$ NPs), serving as an efficient thermocatalyst, are coated on a carbon fiber fabric (Bi$_2$Te$_3$@CFF) to develop a thermocatalytic filter with antibacterial function. Temperature difference induced H$_2$O$_2$ generation by thermocatalysts results in the oxidative damage of bacteria, which makes thermocatalysts highly promising for disinfection applications. Antibacterial activity as high as 95% is achieved only by the treatment of low-temperature difference cycles. The current work highlights the horizon-shifting impacts of thermoelectric materials for real-time purification and antibacterial applications.

[1] Institute of Biomedical Engineering, National Tsing Hua University, Hsinchu 30013, Taiwan. [2] Institute of NanoEngineering and Microsystems, National Tsing Hua University, Hsinchu 30013, Taiwan. [3] Cardiovascular Center, National Taiwan University Hospital, Hsinchu Branch, Hsinchu 30059, Taiwan. [4] College of Medicine, National Taiwan University, Taipei 10051, Taiwan. [5] Institute of Cellular and System Medicine, National Health Research Institute, Zhunan 35053, Taiwan. [6] Department of Orthopaedic Surgery, Spine Section, Chang Gung Memorial Hospital, Taoyuan 33305, Taiwan. [7] Department of Power Mechanical Engineering, National Tsing Hua University, Hsinchu 30013, Taiwan. [8] Frontier Research Center on Fundamental and Applied Sciences of Matters, National Tsing Hua University, Hsinchu 30013, Taiwan. [9] These authors contributed equally: Yu-Jiung Lin, Imran Khan. ✉email: linzh@mx.nthu.edu.tw

The generation of reactive oxygen species (ROS) is the backbone of advanced oxidation processes leading to widespread applications such as organic pollutant degradation[1,2], cancer therapy[3–5], and disinfection[6–8]. Among all different types of ROS, hydrogen peroxide ($H_2O_2$) is more stable than others[9]. As an environmentally friendly chemical, $H_2O_2$ not only plays an important role in various organic syntheses[10–12] but can also be used as a bleaching agent and fuel in industry[13,14]. Conventionally, $H_2O_2$ is produced by the anthraquinone method, which involves some serious limitations, such as complicated reaction steps, where the products and byproducts are hardly separated[15,16]. Recently, photocatalysis and piezocatalysis have emerged as promising alternatives to produce ROS. In particular, the utilization of renewable energies to generate ROS has been widely investigated in recent years instead of traditional chemical methods[17–20]. Light irradiation and mechanical vibration provide the necessary stimuli in photocatalysis and piezocatalysis, respectively. Activated by external triggers, the electron-hole pairs are separated in photo/piezocatalysts and promote surface electrochemical reactions to generate ROS[21]. However, photocatalytic reactions are greatly affected, where suspended matter in wastewater causes low solution transmittance and thus restricts its quantum efficiency below 10%[22,23]. Moreover, in practical applications, round-the-clock performance is also hindered in photocatalysis due to the unavailability of sunlight[24]. On the other hand, the unavailability of suitable mechanical forces in nature creates a serious bottleneck for the realization of effective piezocatalytic performance. Hence, it is imperative to design efficient catalysts that can be exploited in an uninterrupted manner for ROS generation.

Temperature is one of the most important environmental factors, and a temperature difference exists almost everywhere in our living environment. The discovery of thermoelectric materials has paved the path to harness this waste heat or temperature difference for various applications, such as energy convertors and sensors[25,26]. In general, thermoelectric materials can convert thermal energy to electrical energy by means of the temperature difference-induced separation of positive and negative charges[27]. Temperature difference-induced charge separation and consequent voltage generation inside a thermoelectric material can also be expected to realize ROS generation and the corresponding catalytic activity[28,29]. Thermoelectric materials that can be used as catalysts are called thermocatalysts and are endowed with multiple advantages over traditional photo/piezocatalysts. Unlike photocatalysts or piezocatalysts, thermocatalysts show great promise towards the uninterrupted generation of ROS, as a temperature difference always exists in nature. Moreover, temperature difference-induced charge separation in thermocatalysts is beneficial for prohibiting electron-hole recombination, which further ensures higher ROS generation. However, despite these striking advantages, the investigation of ROS generation by thermocatalysts remains almost unexplored in the literature. Since the thermoelectric voltage under a particular temperature difference plays the key role in influencing the thermoelectric effect, a thermoelectric material whose Seebeck coefficient is high should be selected for achieving efficient thermocatalytic performance. Near room temperature, bismuth telluride ($Bi_2Te_3$) has a higher Seebeck coefficient[30–32] than other common thermoelectric materials such as antimony telluride ($Sb_2Te_3$) and lead telluride (PbTe) due to its excellent physical properties and distinctive electronic structure[33].

In this work, we demonstrate $H_2O_2$ generation and the consequent antibacterial activity of selected thermoelectric materials such as $Bi_2Te_3$, $Sb_2Te_3$, and PbTe. Since the early 1800s, $H_2O_2$ has been used as a disinfectant owing to its high oxidative and biocidal efficiency[34]. The thermocatalytic efficiency and subsequent $H_2O_2$ generation were further enhanced 20 times by narrowing down the $Bi_2Te_3$ size. For convenient applications, $Bi_2Te_3$ nanoplates (NPs) were coated on a carbon fiber fabric ($Bi_2Te_3$@CFF), and for simulating real-life conditions, $Bi_2Te_3$@CFF was utilized as a thermocatalytic filter under an indoor temperature difference. $Bi_2Te_3$@CFF exhibits significant antibacterial activity with either a positive or negative temperature difference and shows good durability as the antibacterial activity remains intact for more than one month. The concepts and results presented in this paper strongly highlight the bright prospects of thermocatalysts for $H_2O_2$ generation and environmental disinfection applications.

## Results

**Thermocatalytic performance of bulk thermoelectric materials.** The concept of the thermocatalytic generation of $H_2O_2$ is illustrated in Fig. 1a. $Bi_2Te_3$ is a narrow bandgap semiconductor with a bandgap of ~0.2 eV, and the conduction band potential of $Bi_2Te_3$ is more negative than the redox potential of $O_2/.O_2^-$. In thermal equilibrium, free charges present on the crystal surface quickly exhaust before reacting with the contaminant solution because of the large potential difference between the conduction band and the redox potential of $O_2/.O_2^-$. As a result, no significant catalytic activity is noticed. However, with the application of a temperature difference, negative charges rush from the hot side to the cold side of the material and produce a potential difference between the hot and cold ends. As a result of this thermoelectric potential, the band energy decreases at the positive potential side and increases at the negative potential side. To follow this variation in energy, both the valence band and conduction band tilt across the material and the conduction band comes very close to the redox potential for generating superoxide ($.O_2^-$) radicals. Consequently, electrons from the conduction band can easily migrate to the solution and produce $H_2O_2$ afterward via the reaction. $O_2^- + e^- + 2H^+ \rightarrow H_2O_2$. However, the thermocatalytic property can be exclusively observed in thermoelectric materials, which are capable of producing electron-hole separation under an applied temperature difference. In previous studies, thermoelectric materials have been deployed to realize thermoelectric nanogenerators to harvest thermal energy from the environment and convert it into electrical energy[35,36]. In a similar way, it is expected that the temperature difference-induced potential of thermoelectric materials can be utilized for realizing efficient thermocatalysts.

Therefore, as an attempt to verify this concept, three kinds of commercially available bulk thermoelectric materials, $Bi_2Te_3$, $Sb_2Te_3$, and PbTe, (purchased from Alfa Aeser) were used for the thermocatalytic reactions (Fig. 1b). From the figure, it is evident that all thermoelectric materials can produce $H_2O_2$ under a temperature difference of 20 K. In contrast, the common photocatalyst titanium dioxide ($TiO_2$) cannot produce $H_2O_2$ under the same conditions. The obtained results vividly verify the potential of the thermoelectric materials for using as thermocatalysts. Furthermore, the amount of $H_2O_2$ generated in the thermocatalytic reaction is strongly correlated with the Seebeck coefficient[37] and the dimension of the thermoelectric materials. The Seebeck coefficient of both $Sb_2Te_3$ and PbTe is much smaller than $Bi_2Te_3$ in the low temperature range[30–32], resulting in the highest amount of $H_2O_2$ production in the case of $Bi_2Te_3$. As any catalytic reaction is accelerated with a large surface area, the small size of $Bi_2Te_3$ is also responsible for demonstrating superior thermocatalytic activity as compared to $Sb_2Te_3$ and PbTe (Supplementary Fig. 1). It was also noticed that as the

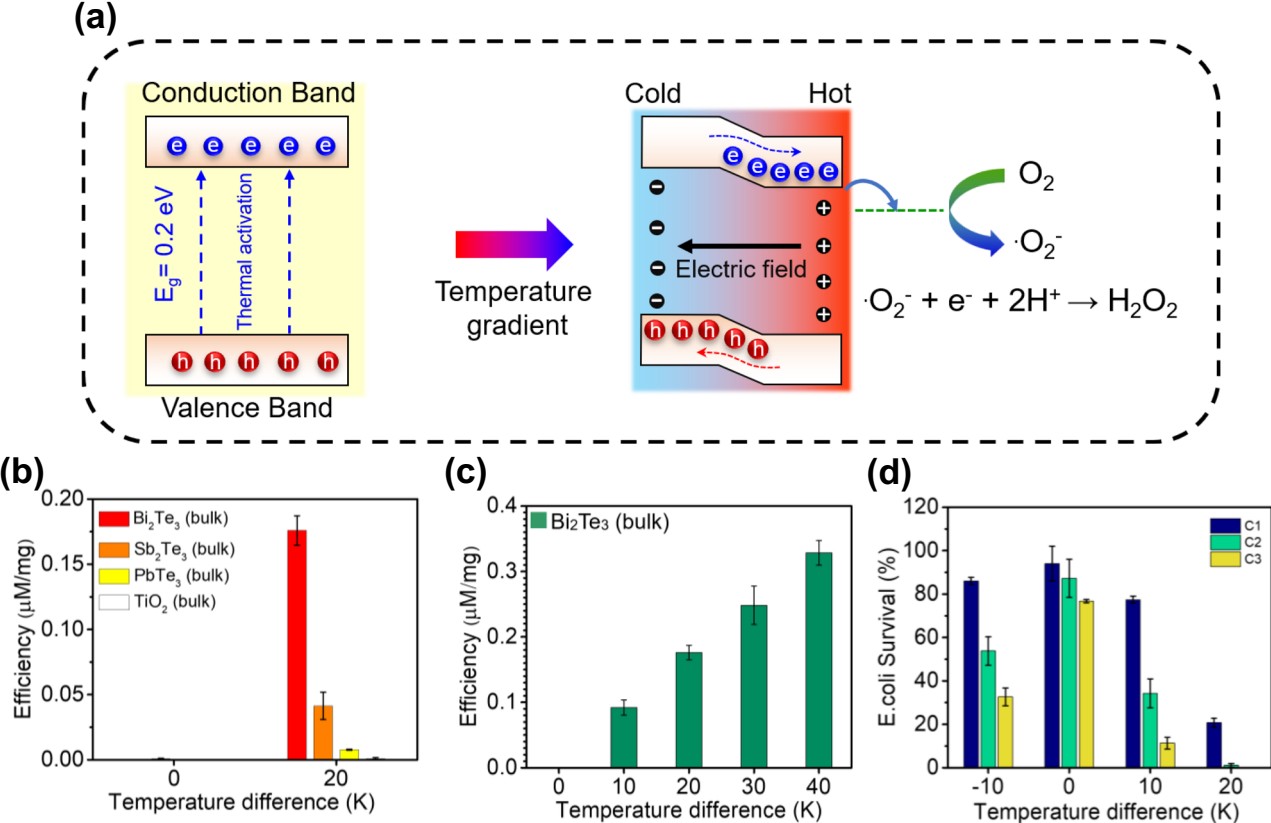

**Fig. 1 Thermocatalytic H$_2$O$_2$ generation by different bulk thermoelectric materials. a** Schematic illustration of the thermocatalysis mechanism for H$_2$O$_2$ generation under a temperature gradient created due to applied temperature difference. Bi$_2$Te$_3$ is selected as a model thermoelectric material to explain the catalytic reaction. **b** H$_2$O$_2$ generation efficiency by three kinds of thermoelectric materials (Bi$_2$Te$_3$, Sb$_2$Te$_3$, and PbTe) and the typical photocatalyst TiO$_2$ at ambient temperature and under a temperature difference of 20 K. **c** H$_2$O$_2$ generation efficiency by bulk Bi$_2$Te$_3$ (50 mg) under different temperature differences. **d** Disinfection performance of bulk Bi$_2$Te$_3$ (50 mg) under different applied temperature differences.

temperature difference was increased to 40 K, the efficiency of H$_2$O$_2$ generation by bulk Bi$_2$Te$_3$ reached approximately 0.34 μM/mg (Fig. 1c). The H$_2$O$_2$ generated by bulk Bi$_2$Te$_3$ under different temperature differences was further used to perform the bacterial disinfection experiments (Fig. 1d). In particular, the bacterial solution was treated with bulk Bi$_2$Te$_3$ under different thermal cycles to observe its disinfection performance. As the number of thermal cycles increases from 1 to 3, the disinfection performance of bulk Bi$_2$Te$_3$ increases gradually. This result also indicates that the disinfection performance has a positive correlation with the amount of H$_2$O$_2$ generated. In other words, higher H$_2$O$_2$ production caused by a higher temperature difference results in a better thermocatalytic effect and consequently a higher disinfection performance.

Control experiments also reveal that the presence of thermocatalysts and a temperature difference are both required for achieving disinfection performance (Supplementary Fig. 2). Moreover, it is also established that superoxide radicals are generated during the thermocatalytic reaction, which facilitates H$_2$O$_2$ generation (Supplementary Fig. 3). Hence, it is conclusively verified that the obtained disinfection performance is a consequence of the in-situ generation of H$_2$O$_2$ by the thermocatalyst under an applied temperature difference. It is also interesting to note that like other conventional catalysts, thermocatalysts also do not undergo any change in weight during H$_2$O$_2$ generation under multiple thermal cycles (Supplementary Fig. 4). Furthermore, when the temperature difference became negligible, no H$_2$O$_2$ was produced, which proved that

H$_2$O$_2$ generation by the thermocatalyst was influenced by the applied temperature difference only (Supplementary Fig. 5).

**Characterization and thermocatalytic activity of Bi$_2$Te$_3$ NPs.** In an attempt to boost the thermocatalytic efficiency, nanometer-sized Bi$_2$Te$_3$ was synthesized via a wet chemical route, and the as-synthesized nanomaterials were characterized by various techniques. The FESEM image shown in Fig. 2a clearly indicates the formation of uniform hexagonal nanoplate-like nanomaterials. Moreover, AFM images also confirm that the thickness of the synthesized Bi$_2$Te$_3$ NPs is ~50 nm (Supplementary Fig. 6a and b). The uniform spatial dispersion of Bi and Te in a single NP is also confirmed from the elemental mapping obtained from EDX analysis (Fig. 2b). The synthesized NPs are highly crystalline in nature, which becomes evident from the HRTEM image exhibiting well-resolved lattice fringes with an interatomic spacing of 0.22 nm corresponding to the (110) planes of Bi$_2$Te$_3$ (Fig. 2b). The obtained hexagonal nanoplates also exhibit high phase purity, which is evidenced from their XRD pattern, as shown in Fig. 2c. All the XRD peaks are well-matched with JCPDS card no. 15-0863, corresponding to rhombohedral structured Bi$_2$Te$_3$ nanocrystals, thereby discarding any possibility of having an impure phase in the synthesized nanomaterials. Similar to their bulk counterpart, Bi$_2$Te$_3$ NPs also exhibit a thermocatalytic effect that can produce H$_2$O$_2$ in the presence of an applied temperature difference (Fig. 2d). Bi$_2$Te$_3$ NPs (5 mg) can generate up to 30 μM H$_2$O$_2$ with a temperature difference of 30 K. It is noteworthy that

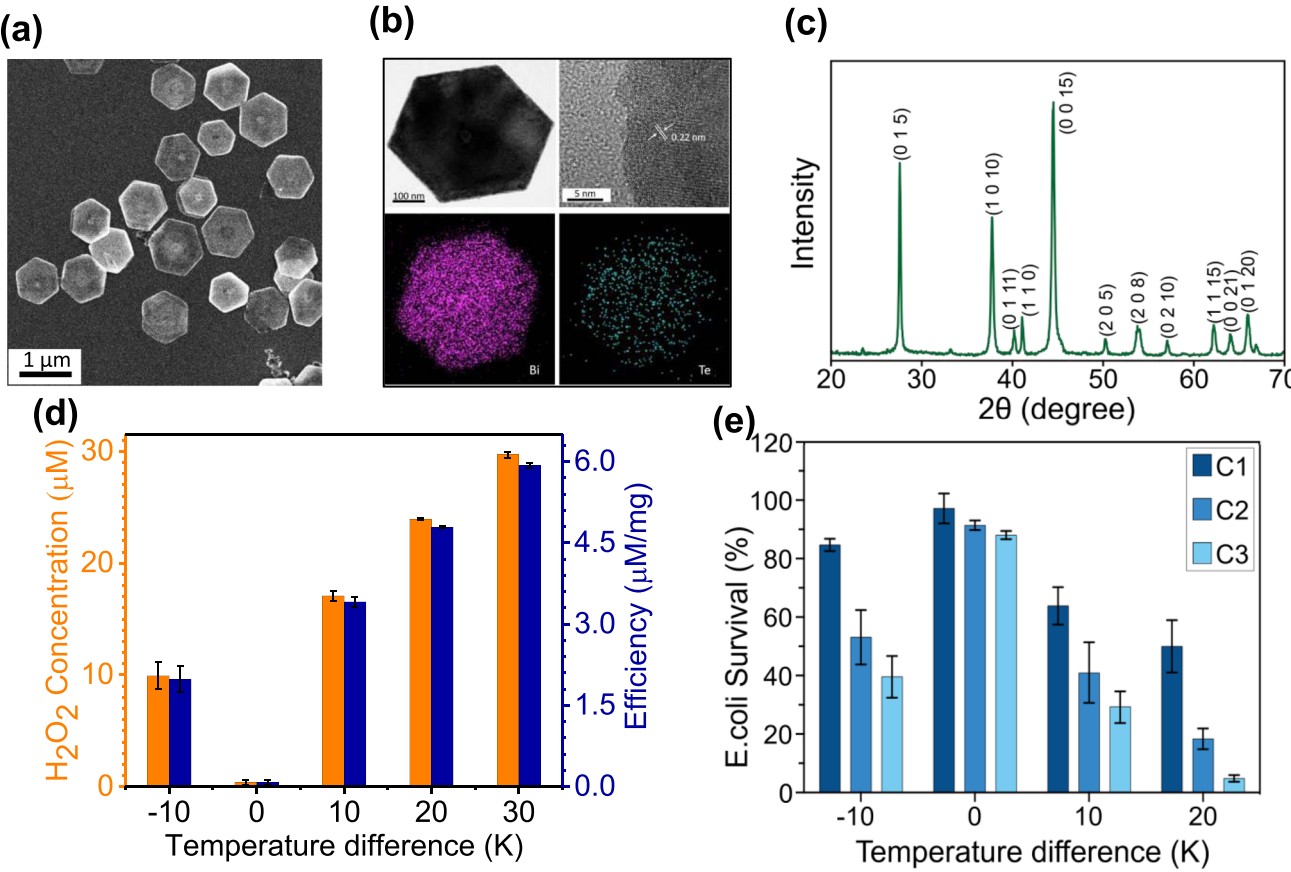

**Fig. 2 Characterization of Bi$_2$Te$_3$ NPs for H$_2$O$_2$ generation and disinfection performance. a** SEM image demonstrating the uniform hexagonal nanoplate-like morphology of the synthesized nanomaterials. **b** HRTEM image and elemental mapping images revealing the well-resolved lattice fringes corresponding to the (110) plane and homogeneous distribution of Bi and Te in the as-prepared Bi$_2$Te$_3$ NPs, respectively. **c** XRD pattern of the as-prepared Bi$_2$Te$_3$ NPs. **d** H$_2$O$_2$ generation by the as-prepared Bi$_2$Te$_3$ NPs under different temperature differences. **e** Disinfection performance of the as-prepared Bi$_2$Te$_3$ NPs under various temperature differences and thermal cycles.

regardless of whether the temperature difference is positive or negative, H$_2$O$_2$ can still be generated.

Since the Seebeck coefficient of a thermoelectric material plays the pivotal role in controlling the thermocatalytic activity, the thermoelectric voltage and corresponding Seebeck coefficient of Bi$_2$Te$_3$ NPs are also estimated (Fig. 3). The thermoelectric device fabricated by Bi$_2$Te$_3$ NPs was subjected to a series of temperature difference by adjusting the temperature (40 to 100 °C) of a hot plate. When the temperature of the hot end of the device is increased from 40 to 100 °C, the generated thermoelectric voltage is also enhanced from 0.72 mV to 19.83 mV (Fig. 3a, b). The actual temperature difference created across the device is represented by the temperature distribution images as shown in Fig. 3a. The developed thermoelectric voltage is a consequence of temperature difference induced carrier migration within the Bi$_2$Te$_3$ NPs. The obtained thermoelectric voltages (ΔV) are plotted as a function of actual temperature difference (ΔT) and the slope obtained from the linear fitting of ΔV vs. ΔT curve denotes the Seebeck coefficient (S) (Fig. 3c). The synthesized Bi$_2$Te$_3$ NPs exhibit a high Seebeck coefficient of ~ 497 µV/K which strongly validates the potential of Bi$_2$Te$_3$ NPs to act as efficient thermocatalyst. It is noteworthy that when one end of the thermoelectric device is cooled keeping the other end at room temperature, the charge carriers within Bi$_2$Te$_3$ NPs move to a reverse direction and consequently the opposite output voltage signals are obtained (Supplementary Fig. 7). Moreover, it is also important to note that obtained thermoelectric voltage for applied +10 K temperature difference is higher (0.45 mV) than the

generated thermoelectric voltage (0.19 mV) for −10 K temperature difference (Supplementary Fig. 7b). The impact of this phenomenon is vividly reflected from the H$_2$O$_2$ generation performance of Bi$_2$Te$_3$ NPs at different temperature differences as shown in Fig. 2d. The amount of H$_2$O$_2$ generated under a −10 K temperature difference (8.5 µM) is lower than that generated under a +10 K temperature difference (18 µM). Obtained results strongly signify that the generated thermoelectric voltage of a thermocatalyst plays a crucial role in controlling surface electrochemical reaction and consequent catalytic activity.

It is also interesting to note that H$_2$O$_2$ generation efficiency is significantly higher in the case of cyclic heating condition than the continuous heating condition (Supplementary Fig. 8). Since, for nanomaterials it is difficult to maintain the temperature difference for a long time, in the current work cyclic heating operation was used in which the thermocatalyst restores their initial temperature when they are kept back at room temperature after each heating step. The synthesized Bi$_2$Te$_3$ NPs were also used in disinfection experiments, and the results are shown in Fig. 2e. From the figure, it is evident that significant antibacterial activity was achieved under a temperature difference of 20 K, where only 5% of bacteria survived. However, control experiments revealed that bulk Bi$_2$Te$_3$ (5 mg) exhibited very little antibacterial activity under the same experimental conditions as Bi$_2$Te$_3$ NPs (Supplementary Fig. 9).

Temperature difference-induced voltage generation in the thermoelectric materials was further confirmed from the surface potential analysis carried out by Kelvin probe force microscopy

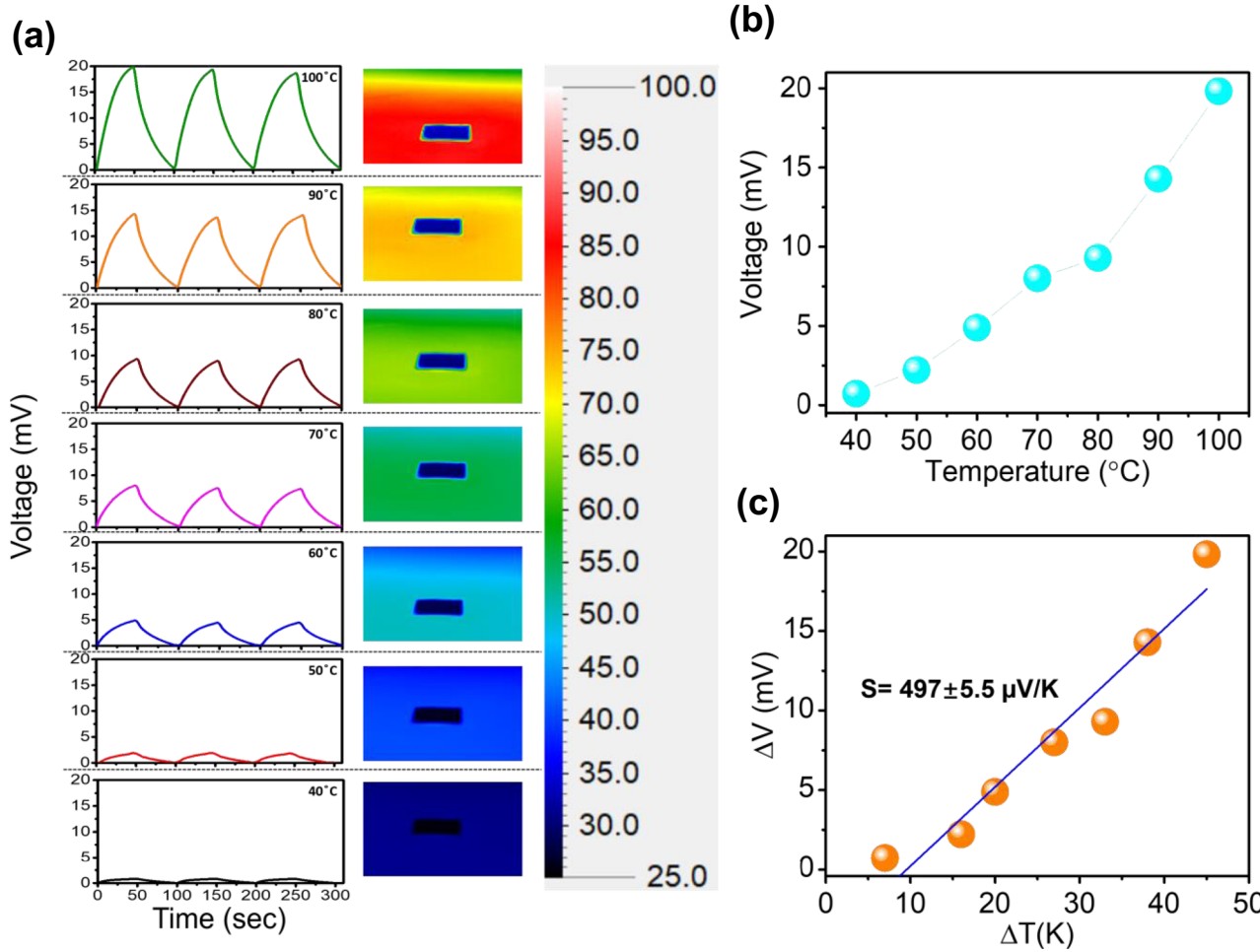

**Fig. 3 Thermoelectric characterizations of Bi₂Te₃ NPs. a** Demonstration of thermoelectric voltage generation by Bi₂Te₃ NPs at different heating temperatures. Corresponding actual temperature differences created across the device are represented by the temperature distribution images recorded by IR camera. **b** Variation of thermoelectric voltage with heating temperature ranging from 40 to 100 °C. **c** Slope of the linear fitting of ΔV vs. ΔT curve indicates the Seebeck coefficient.

(KPFM) incorporated with a thermal stage to heat the samples (Fig. 4a, b). While the surface of Bi₂Te₃ NPs does not show any surface potential in thermal equilibrium, with the application of a temperature of 60 °C, a ~280 mV response voltage is observed. When an applied temperature difference is imposed across the Bi₂Te₃ NPs, charge carriers with opposite polarity are separated from the surfaces, and as a result, the Bi₂Te₃ NPs gain more surface charges. Under this condition, to maintain the Fermi level alignment, the contact potential difference (CPD) between the KPFM tip and sample increases compared to thermal equilibrium conditions. A higher contact potential difference also indicates a reduction in the work function of Bi₂Te₃ under the applied temperature[38]. The observed output voltage is a direct consequence of temperature difference-induced electron-hole separation in Bi₂Te₃. Moreover, the generation of H₂O₂ by Bi₂Te₃ was tested under different environmental conditions, including O₂, N₂, and air. It was found that the N₂-filled environment led to the generation of the lowest amount of H₂O₂, indicating that superoxide radical derivation from dissolved O₂ was the key factor in H₂O₂ generation (Fig. 4c). It is also found that the increase in temperature difference boosts the thermocatalytic generation of superoxide radicals which is evidenced from Supplementary Fig. 10. Moreover, in order to prove that generation of superoxide radical is thermodynamically possible, the band edge position of Bi₂Te₃ NPs was determined with the

help of high-resolution valence band XPS (Supplementary Fig. 11). The valence band maxima (VBM) is determined from the binding energy position at which the electronic density of states become zero. From Supplementary Fig. 11 it is clear that the VBM position of Bi₂Te₃ NPs is located at −0.41 eV which is in close agreement with the VBM position of Bi₂Te₃ reported previously[39]. Conduction band minima (CBM) positions are calculated from the equation $E_{CB} = E_{VB} - E_g$ ($E_g$, $E_{CB}$ and $E_{VB}$ indicate the bandgap energy, conduction band potential, and valence band potential). Considering the bandgap of Bi₂Te₃ as 0.2 eV[40], the CBM position is calculated to be −0.61 eV. The obtained conduction band potential is more negative than the redox potential of O₂/·O₂⁻ (−0.33 eV vs. NHE)[41,42]. The more negative conduction band potential readily implies that formation ·O₂⁻ radical and consequent H₂O₂ generation is thermodynamically possible. Furthermore, the efficiency of H₂O₂ generation by the Bi₂Te₃ NPs was compared to that of bulk Bi₂Te₃ (Fig. 4d), revealing that the amount of H₂O₂ generated by Bi₂Te₃ NPs was at least 20 times higher than the amount generated by bulk Bi₂Te₃. Since catalysis is a surface phenomenon, the high surface area of the catalysts can effectively boost the surface chemical reactions by increasing the active catalytic sites. The 2D plate-like structure of Bi₂Te₃ NPs facilitates higher surface area than bulk Bi₂Te₃ which is responsible for this superior thermocatalytic acivity. The nanostructuring approach for realizing high surface

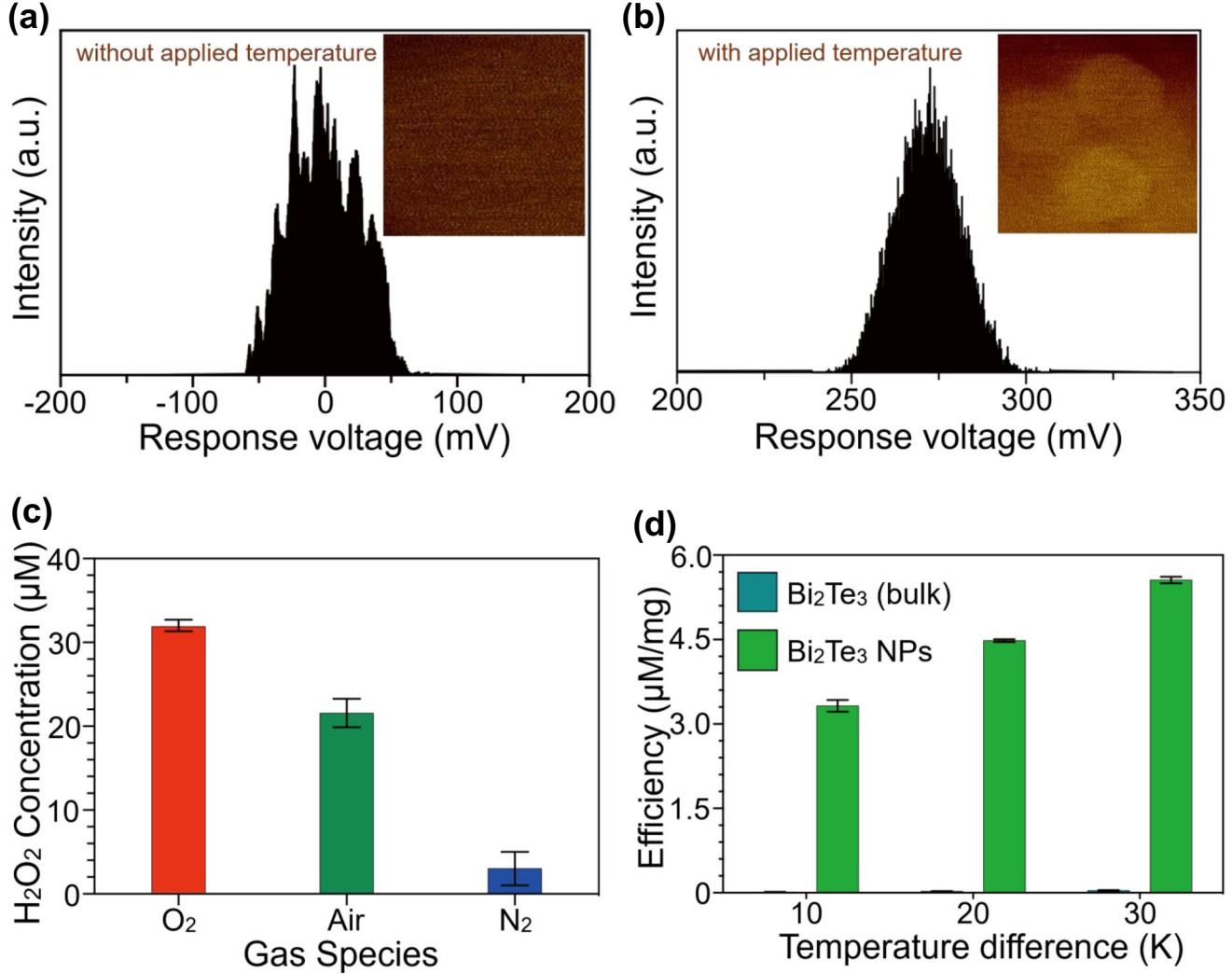

**Fig. 4 Temperature-induced surface potential evaluation and H₂O₂ generation.** Thermoelectric voltage generation by the as-prepared $Bi_2Te_3$ NPs measured by KPFM equipped with a thermal stage **a** at room temperature and **b** at 60 °C. **c** $H_2O_2$ generation by the as-prepared $Bi_2Te_3$ NPs in different ambient conditions of $O_2$, air, and $N_2$ under a temperature difference of 20 K. **d** The efficiency of $H_2O_2$ generation by the as-prepared $Bi_2Te_3$ NPs compared to bulk $Bi_2Te_3$.

area and efficient catalytic activity are already widely reported in the literature[17,43]. Moreover, since $Bi_2Te_3$ is a narrow bandgap semiconductor, it is necessary to find out if there is any contribution from photocatalysis in the thermocatalytic activities shown by $Bi_2Te_3$ NPs. In presence of ambient light, $Bi_2Te_3$ NPs exhibit almost similar catalytic performance as compared to dark (Supplementary Fig. 12). Actually, for the thermocatalysts, temperature difference induced thermoelectric voltage generation and consequent band bending is primarily responsible for reducing the energy difference between band energy and redox potential which is eventually not possible for a photocatalyst. Hence the obtained thermocatalytic performance remains almost unaffected from the interference of photocatalysis. These phenomena not only indicate that ambient light has negligible effect in the thermocatalytic activity offered by $Bi_2Te_3$ but also support the practical applicability of the thermocatalysts in the real circumstances.

**$Bi_2Te_3$@CFF for antibacterial filter applications**. The excellent disinfection performance of $Bi_2Te_3$ NPs was further employed to design an antibacterial filter with potential applications in our

daily life. The antibacterial filter was fabricated by coating the $Bi_2Te_3$ NPs on a CFF, as illustrated in Fig. 5a. Here, CFF was chosen as an antibacterial filter because of its potential applications in textile industries. The uniform deposition of $Bi_2Te_3$ NPs over CFF is clearly visible from the FESEM images shown in Fig. 6a. Moreover, it is also interesting to note that the $Bi_2Te_3$ NPs retain their hexagonal shape even after dip-coating on the CFF. As expected, the developed antibacterial filter is capable of producing $H_2O_2$ by a thermocatalytic effect when it is exposed to a temperature difference (Supplementary Fig. 13). However, bare CFF does not exhibit any $H_2O_2$ generation under the applied temperature difference.

The temperature difference needed for $H_2O_2$ generation can also be harnessed from the surrounding environment. Cold air and hot air coming from a cooling fan and a hairdryer, respectively, were directed onto the front side of the $Bi_2Te_3$@CFF to produce a temperature difference through the filter. The produced temperature difference across the $Bi_2Te_3$@CFF filter is clearly evident from the real temperature distribution curve obtained from the thermal image, as shown in Fig. 5b, c. Since cold air was blown onto the front side, the temperature at the front side of the filter decreased to approximately 25 °C (Fig. 5b).

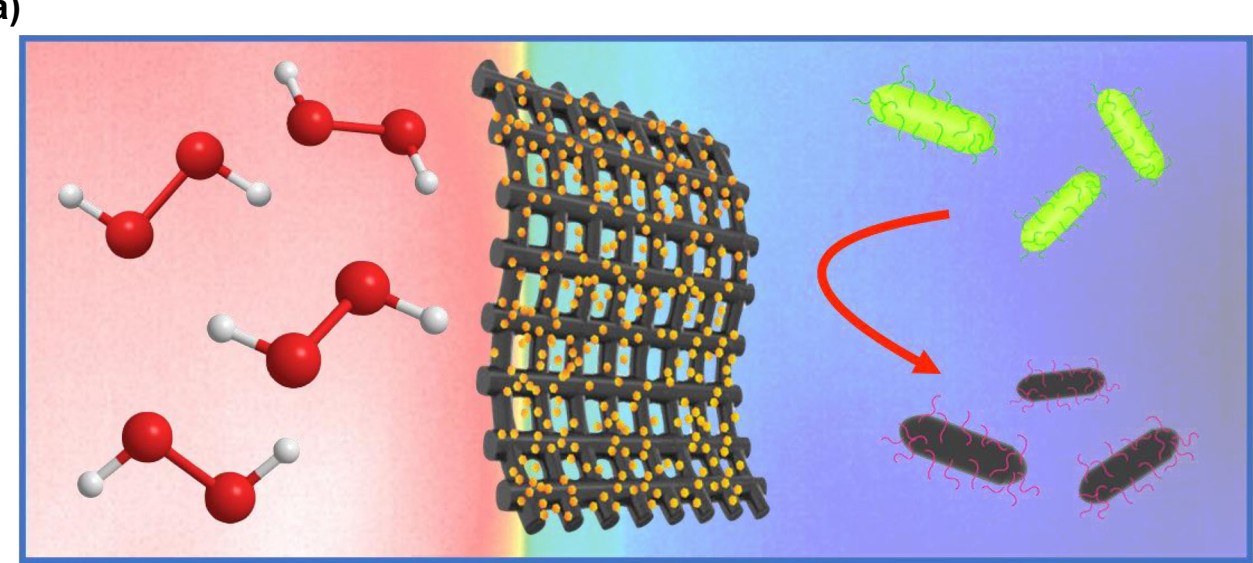

**Fig. 5 Disinfection performance of thermocatalytic filter with cooling and heating effect. a** Schematic illustration of the thermocatalytic filter developed by coating the as-prepared Bi₂Te₃ NPs on a commercial CFF. **b**, **c** Temperature profile of the thermocatalytic filter captured by an IR camera during exposure to cold air and hot air, respectively. The top insets show the illustration of creating a temperature difference by driving cold air and hot air onto the thermocatalytic filter. Bottom insets show the 3D temperature distribution images of the thermocatalytic filter. **d**, **e** Disinfection performance of Bi₂Te₃@CFF driven by cold air and hot air, respectively. Control experiments using the pure CFF indicate that no obvious disinfection performance was observed without the as-prepared Bi₂Te₃ NPs.

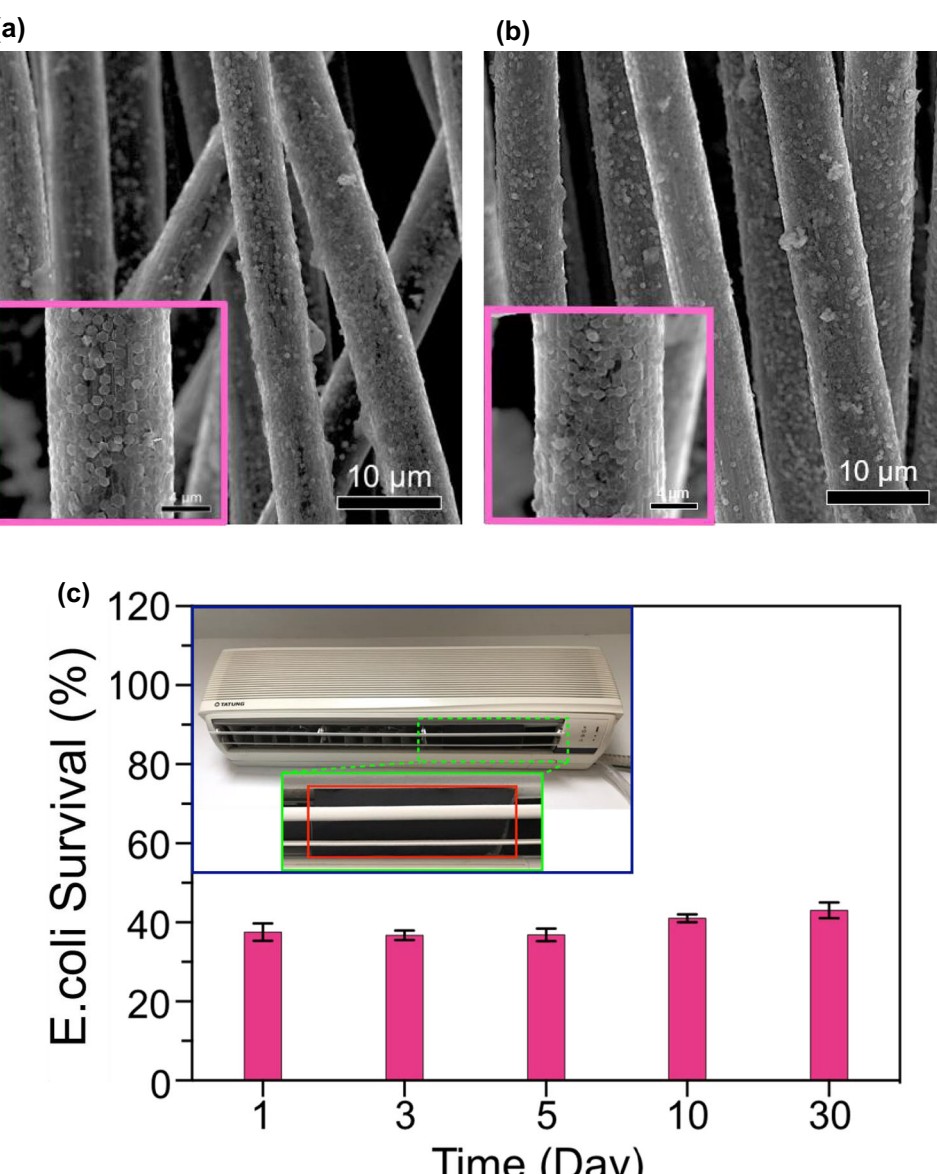

**Fig. 6 Real-time application of antibacterial filter and its reusability test. a** SEM image of Bi$_2$Te$_3$@CFF after dip-coating. The inset shows the high magnification view of Bi$_2$Te$_3$@CFF representing uniform deposition of Bi$_2$Te$_3$ hexagonal nanoplates on the CFF. **b** SEM image of Bi$_2$Te$_3$@CFF after 30 days of use. **c** Reusability test demonstrating the disinfection performance of Bi$_2$Te$_3$@CFF for 30 days. The inset shows a digital photograph indicating the installation of Bi$_2$Te$_3$@CFF in the air conditioner.

On the other hand, the temperature on the front side increases to more than 45 °C when hot air was blown from a hairdryer (Fig. 5c). The disinfection performance of Bi$_2$Te$_3$@CFF was also probed under cold air or hot air treatment and compared with that of a commercially available unmodified CFF (Fig. 5d, e). The commercial unmodified CFF does not exhibit any antibacterial activity even upon the application of a temperature difference which is fully consistent with the poor H$_2$O$_2$ generation performance shown by bare CFF as represented in Supplementary Fig. 13. However, significant disinfection performance was achieved for Bi$_2$Te$_3$@CFF under the treatment of both cold and hot air. It is also interesting to note that since the achieved temperature difference across the Bi$_2$Te$_3$@CFF is higher for hot air treatment, the bacterial survivability (~28%) is also less in this condition as compared to cooling fan experiment. Moreover, it was also noticed that without the application of cold air or hot air, no antibacterial activity was observed, which proves the necessity

of the thermocatalyst for effective antibacterial performance in the presence of a temperature difference.

For further practical applications of the thermocatalyst as a disinfectant of the indoor local environment, a large-sized Bi$_2$Te$_3$@CFF was fabricated for utilization as a domestic antibacterial filter. An air conditioner (AC) is a commonly used machine to control the temperature and humidity in indoor surroundings. Herein, we used an air conditioner as a platform to demonstrate the concept of indoor air purification technology. The antibacterial filter was installed in the air conditioner in such a way that it could be easily removed or changed. The temperature inside the air conditioner was lower than the air in the surroundings, creating a temperature difference to trigger the thermocatalytic reaction. Since the air conditioner was set at 17 °C, it can be concluded that the temperature difference across the Bi$_2$Te$_3$@CFF was similar to that in the case of cooling fan experiment. This fact is also clear from the disinfection results

which reveal that thermocatalytic reaction at air conditioner also leads to ~ 60% bacterial degradation as in the case of cooling fan experiment. A comparison of the SEM image shown in Fig. 6a, b clearly indicates that Bi$_2$Te$_3$@CFF does not undergo any degradation even after 30 days of operation. XRD and Raman spectra of the Bi$_2$Te$_3$@CFF (Supplementary Fig. 14) also indicate that even after 30 days of thermocatalytic operation under air conditioner, the antibacterial filter retains their initial spectral characteristics. As a result of these, the antibacterial activity of the designed filter can be well retained even after 30 days of use (Fig. 6c). The obtained results strongly indicate the long-term stability and robustness of the designed antibacterial filter for practical disinfection applications.

## Discussion

In summary, Bi$_2$Te$_3$-based thermocatalysts have been designed to demonstrate temperature difference-induced H$_2$O$_2$ generation in conjunction with bacterial disinfection. In particular, Bi$_2$Te$_3$ NPs were synthesized by a simple wet chemical route and deployed under a positive or negative temperature difference to realize surface electrochemical reaction-mediated ROS generation. As the temperature difference was increased to 30 K, the amount of H$_2$O$_2$ generated by the Bi$_2$Te$_3$ NPs was approximately 30 μM, which is ~20 times higher than that generated by bulk Bi$_2$Te$_3$. The thermoelectric voltage generated by Bi$_2$Te$_3$ NPs under a temperature difference is identified to play a decisive role in controlling the thermocatalytic ROS generation process. The concept of using thermoelectric materials for ROS generation was also verified by testing the thermocatalytic performance of other thermoelectric materials, such as Sb$_2$Te$_3$ and PbTe. Moreover, the Bi$_2$Te$_3$ NPs exhibit excellent disinfection performance by achieving 95% antibacterial activity (5% E. coli survival) after three thermal cycles. For the effective utilization of the developed catalysts in daily life applications, Bi$_2$Te$_3$ NPs were coated on a CFF and used as an antibacterial filter in a domestic air conditioner. As long as a temperature difference exists, the thermocatalyst can generate ROS to kill the bacteria. As a whole, the concepts presented here highly promote the merits of thermocatalysts for round-the-clock ROS generation that can open a new direction towards sustainable environmental remediation applications.

## Methods

**Synthesis of Bi$_2$Te$_3$ NPs**. First, a stock solution was prepared by dissolving 0.8 g sodium hydroxide in 10 mL ethylene glycol at 45 °C in a water bath. Then, 0.1 g bismuth nitrate pentahydrate, 0.067 g sodium telluride, and 0.235 g poly-vinylpyrrolidone (PVP) were loaded into a 25 mL three-neck flask, to which 10 mL stock solution was added. The mixture was stirred for 10 min at room temperature. After that, the three-neck flask was placed in the water bath at 45 °C for 20 min. Until the precursors were dissolved in the solution, a three-neck flask was kept in the oil bath and stirred for 3 h at 190 °C. After the reaction, 30 mL of isopropyl alcohol and 10 ml of acetone were added to the solution, and the mixture was centrifuged at 6700 × $g$ for 10 min. The supernatant was discarded, and the process was repeated 3 times. Finally, the filtered Bi$_2$Te$_3$ NPs were redispersed in 30 mL of isopropyl alcohol and were used for further experiments.

**Preparation of Bi$_2$Te$_3$@CFF**. To clean the surface of the carbon fiber fabric, it was soaked in acetone, isopropanol, and DI water for 5 min, respectively. Then, the carbon fiber fabric was cut into a size of 1 × 1 cm and dipped in 100 μL of Bi$_2$Te$_3$ NP (1 mM) solution. Finally, the as-prepared carbon fiber was dried in a hot air oven to completely remove the water.

**Characterization**. X-ray diffraction (XRD, Rigaku TTRAX III) was employed to investigate the structural properties of the prepared nanomaterials. For XRD analysis, the Bi$_2$Te$_3$ NPs were coated on a glass slide and placed on a hot plate to remove water. The morphology of the Bi$_2$Te$_3$ NPs was analyzed by using scanning probe microscopy (SPM, Bruker). The morphology of the Bi$_2$Te$_3$@CFFs was analyzed via field emission scanning electron microscopy (FESEM, JEOL JSM-7600F). For FESEM analysis, the Bi$_2$Te$_3$ NPs were drop-coated on a Si wafer and placed in a 60 °C incubator overnight. Cs-corrected STEM (JEOL ARM200F) and energy dispersive spectrometry (EDS, Oxford INCA) were used to analyze the

lattice and elemental distribution of the Bi$_2$Te$_3$ NPs, respectively. Amplitude modulated Kelvin probe force microscopy (AM-KPFM) equipped with a thermal stage was employed to monitor the surface potential distribution of the Bi$_2$Te$_3$ NPs. Raman spectra were recorded by a UniRam Raman spectrometer (ProTrus. Tech. Co. Ltd.) equipped with a 532 nm laser. Valence band XPS (VB-XPS) spectra were collected by X-ray photoelectron spectrometer (ESCALAB 250 XI, Thermo Scientific) having monochromatic Al Kα X-ray source.

**Measurement of Seebeck coefficient**. A device size of 1cm$^2$ was used for the thermoelectric voltage measurement and a hot plate was used to create the temperature difference. For measuring the thermometric voltage below room temperature, a PID controlled cooling stage was used. The temperature difference between the surface of the hot-plate/cooling stage and the top surface of device induces temperature difference across the device and the consequent thermoelectric voltage is generated. The generated thermoelectric voltages were recorded with a low noise voltage preamplifier (SR-560). To determine the actual temperature difference created across the device, the temperature at the surface of the hot-plate (T1) and top of the device (T2) were monitored with the help of IR camera.

**Preparation of E. coli for the disinfection test**. E. coli K12 cells were grown in lysogeny broth (LB) medium. The original E. coli K12 cells were added to LB medium for 16 h at 37 °C in an incubator. After 16 h, the E. coli K12 cells were diluted to an optical density of 0.06 at 670 nm (OD$_{670}$ = 0.06). Then, the bacterial cell suspension was diluted 10 times in 0.85% sodium chloride, which was equal to 2 × 10$^7$ CFU per 1 mL for the antibacterial investigation.

**Thermocatalytic disinfection test**. Thermocatalysis experiments were conducted in a water bath where the thermocatalysts in the contaminant solutions were subjected to 3 thermal cycles denoted as C1, C2, and C3. In each cycle, the materials at first were allowed to react at specific temperature (15/35/45 °C) for 5 min and then were returned to room temperature (RT) for 5 min. Two separate groups of experiments were carried out, one without a thermal cycle (under RT) and the other with thermal cycles. The materials were bulk Bi$_2$Te$_3$ and Bi$_2$Te$_3$ NPs. The masses of the bulk Bi$_2$Te$_3$ and Bi$_2$Te$_3$ NPs were 0.05 g and 0.005 g, respectively. These materials were added to 1 mL bacterial solution (2 × 10$^6$ CFU per 1 mL) for the thermocatalysis experiments.

**Detection of ·O$_2$$^-$ and H$_2$O$_2$ generated by the thermocatalysts**. The generation of superoxide radicals during the thermocatalysis reaction was estimated quantitatively by using XTT (2, 3-bis (2- methoxy-4-nitro-5-sulfophehyl)−2H-tetra-zolium-5-carboxanilide) assay. Typically, aqueous dispersion of bulk and Bi$_2$Te$_3$ NPs were mixed with XTT (50 μM) and then, the solution was subjected to different temperature difference at water bath. After the reaction Bi$_2$Te$_3$ was separated by centrifugation and the absorbance spectra of supernatants were monitored at 470 nm.

Amplex Red reagent with HRP enzyme was used for H$_2$O$_2$ detection. In general, Amplex Red reacts with H$_2$O$_2$ to produce the red-fluorescent oxidation product resorufin. First, two different stock solutions were prepared. One was 0.4 mg Amplex Red powder dissolved in 3.1 mL dimethyl sulfoxide (DMSO), and the other was 0.5 mg of HRP dissolved in phosphate-buffered saline (PBS, pH 5.8). The materials (bulk Bi$_2$Te$_3$, bulk Sb$_2$Te$_3$, bulk PbTe, bulk TiO$_2$, Bi$_2$Te$_3$ NPs, Bi$_2$Te$_3$@CFFs) were first added to a 1 mL sodium chloride (0.85% NaCl) solution, and the solution was then kept in a water bath for 15 min where it was subjected to different temperatures (10, 15, 35, 45, 55, 65 °C). After the temperature treatment, the solution was filtered by a 0.2 μm PVDF membrane filter, and 270 μL of filtrate solution was added to a mixture of 30 μL Amplex Red solution and 3 μL HRP solution. A photoluminescence spectrophotometer (HITACHI F-7000) was used to detect the generated fluorescent product. The samples were excited at 530 nm, and emission spectra were scanned from 560~750 nm.

**Disinfection test by the antibacterial filter**. To demonstrate the performance of Bi$_2$Te$_3$@CFF as an antibacterial filter, the disinfection tests were conducted using a hairdryer and a cooling fan. At first 1 mL of bacterial solution (2 × 10$^6$ CFU/mL) was added to 1 × 1 cm Bi$_2$Te$_3$@CFF and further treated for a total time of 20 min under the temperature difference created by hairdryer and cooling fan. Control experiments were also performed under the same conditions without the temperature difference. Both the treated and untreated Bi$_2$Te$_3$@CFF were immersed in 1 mL of 0.85% sodium chloride solution. Aliquots of 100 μL of bacterial solution were collected from each group and plated on an aseptic plate. The bacterial colonies were counted from the plates after 24 h of incubation at 37 °C. The survival rates were determined by using the formula C/C$_0$ × 100% where C$_0$ is the concentration of the bacteria solution before thermal treatment and C is the remaining concentration of the bacteria after the thermal treatment.

**Reusability test of the antibacterial filter**. This test was conducted by using an air conditioner. Bi$_2$Te$_3$@CFF with dimensions of 8 × 15 cm was mounted on an indoor part of a split air conditioner. The temperature of the air conditioner was set to 17 °C

in order ensure similar range of temperature difference as in case of cooling fan experiment. In this case, the same bacterial concentration ($2 \times 10^6$ CFU/mL) was used to react the $Bi_2Te_3$@CFF as in the case of cooling fan experiment. The treatment was the same as the air for the disinfection test. The treatment was repeated for 30 days.

**Reporting Summary**. Further information on research design is available in the Nature Research Reporting Summary linked to this article.

## Data availability

Source Data are provided with this paper. All the experimental results of the main manuscript and Supplementary Information are available in Figshare repository via the identifier https://figshare.com/s/e2460939d5f21d4aefdc.

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

## Acknowledgements

This work was financially supported from the Young Scholar Fellowship Program by Ministry of Science and Technology in Taiwan (MOST109-2636-E007-013), the Ministry of Education in Taiwan (MOE 107QR00115), the National Tsing Hua University (109Q2501E1), and the National Taiwan University Hospital Hsinchu Branch (109-HCH095).

## Author contributions

Z.-H.L. conceived the research idea and supervised the entire project. Y.-J.L., I.K. designed the experiments and carried out the material synthesis and device fabrication. S.S., Y.-J.L., and I.K. analyzed the results and wrote the manuscript. S.R.B. and C.-C.W. conducted the antibacterial experiments. Biocompatibility tests were carried out by F-C. K. All the authors discussed the results and commented on the manuscript.

## Competing interests

The authors declare no competing interests.
