## [Peer Review File · Nature Communications]

Reviewers' Comments:

Reviewer #1:

Remarks to the Author:

In the paper, the authors present a new mechanism to produce ROS for Hydrogen Peroxide Generation and Bacterial Disinfection. It is a kind of interesting relative to the photocatalyst and piezocatalyst, but the thermocatalyst has already be proposed and used for carbon dioxide hydrogenation (Tuning of Catalytic Activity by Thermoelectric Materials for Carbon Dioxide Hydrogenation, *Advanced Energy Materials*, 2018, 8, 1701430; Thermoelectric avenue by James Gallagher). Having said that, the paper present interest to the reader for ROS, but not enough for the "Thermoelectric avenue". In the revision, the so-tightly related refs should be added. Below are my comments and questions.

What is the mean of "w/o cooling and w/ cooling" in Figure 4

The author demonstrated a practical application for the thermocatalyst in Figure 5. From the text, it is not clear that how did they obtain the E.coli of 40%? Did you do the experiment via cooling fan or a hair dryer, as that in Figure 4?

If yes for the above question (seems yes, because they said the treatment was the same as the air for the disinfection test the), what is the real temperature gradient for the antibacterial filter in the air conditioner, and what is the E.coli values? A real demo is necessary for this concept, because the demo in Figure 4 with a huge temperature gradient of >10 K.

In Figure 5, the two SEM images are presented in order to reveal the reusability of the thermocatalyst. However, it might be not from the same zone, and why the inset of the SEM images are with different size scale, 2 μm and 4 μm ???. The SEM images are absolutely not enough to show the reusability of thermocatalyst, XRD and other characterization are necessary.

Line 193, "...revealing that the amount of H₂O₂ generated by Bi₂Te₃ NPs was at least 20 times higher than the amount generated by bulk Bi₂Te₃." More discussion are necessary to the readers.

In Figure 1, from the thermocatalysis mechanism for H₂O₂ generation, the generation of .O₂- is necessary. The authors should present the evidence for the mechanism. In my opinion, the concentration of .O₂- by different temperature gradient or materials should be provided.

How about the bio-capability of Bi₂Te₃ NPs? Is that health for air condition filter?

In the title, "Highly efficient" was used, related discussion should be addressed.

Reviewer #2:

Remarks to the Author:

In this work, the authors reported Bi₂Te₃ thermocatalyst for hydrogen peroxide generation and bacterial disinfection applications. Though there are some points need to be clarified, it is an interesting work that may finds possible application in the future.

1. In line 126-127, "...reaction is strongly correlated with the figure of merit (zT value) and the dimension of the thermoelectric material. The zT value of PbTe is much smaller than both Bi₂Te₃ and Sb₂Te₃ in the low temperature range." Since zT value is a very important factor to consider for thermocatalyst, the authors provide the zT values of samples (coated on CFF) used in this study.
2. Following the above comment, the underlying mechanism of thermocatalyst is not very clear stated in the present work. The authors use Fig.1a to explain the mechanism. Based on this model, what is important for thermocatalysis should be the thermoelectric voltage. Therefore, the authors should

compare the Seebeck coefficient of their materials since the thermoelectric voltage generated on two ends of their samples is proportional to the Seebeck coefficient, while zT is related to the power output. Besides, the authors mistakenly treat temperature difference as temperature gradient. What the authors measured are actually temperature difference, not gradient and it would be very difficult to accurately measure the gradient in the current work if not impossible. Fortunately, the thermoelectric voltage depends on the temperature difference, not the gradient.

3. To assist the model mechanism proposed in Fig.1a, detailed experiments should be conducted to determine the band edge positions of samples to verify that the H_2O_2 generation reaction is at least thermodynamically possible.

4. The thermoelectric materials are narrow bandgap semiconductors. Are they also photocatalysts for H_2O_2 generation? If yes, how to avoid the interference of photocatalysis during the experiment?

5. In line 136, "As the number of thermal cycles (C1, C2, and C3) increases..." It is better to provide accurate cycle number such as 20, or 30. Besides, there is no experimental details to tell how the bulk materials (Bi_2Te_3 and others) are prepared and tested in Fig.1b-1d.

6. In the experiment section, according the statement "In each cycle, the materials at first were allowed to react at specific temperature (15/35/45) for 5 min and then were returned to room temperature (RT) for 5 min..." It seems that the temperature gradient was obtained by immerse the sample in hot and cold water alternatively. I wonder the temperature difference across the nanoplate would be large enough to have some real effect in catalysis, considering the thickness of Bi_2Te_3 nanoplate is in the nano-range. Besides, the temperature difference will soon diminish due to thermal conduction. It would be more convincing if a constant temperature difference across thermoelectric sample can be maintained.

7. In the disinfection test, the temperature on the back side of the filter in Fig.4b and 4c is supposed to be the ambient temperature, which should be the same. Why is there a large difference in the ambient temperature, which is ~ 34 degree in Fig.4b and ~ 28 degree in Fig.4c? How long was the disinfection test conducted?

8. For air conditioner antibacterial filter experiment. It is better to provide the result of a control experiment using a non-coated CFF filter for comparison. Control experiment using bare CFF sample should also be conducted in Fig.2d for comparison.

9. It is meaningless to show the total H_2O_2 generation in the experiments in Fig.1 and 2. Instead, the H_2O_2 generation per unit mass of thermoelectric material should be shown.

10. Lines 110-112, when the valence and conduction bands are tilted, how can both of them come close to the redox potential for generating superoxide?

Reply to the reviewers' comments

Reviewer #1 (Remarks to the Author):

In the paper, the authors present a new mechanism to produce ROS for Hydrogen Peroxide Generation and Bacterial Disinfection. It is a kind of interesting relative to the photocatalyst and piezocatalyst, but the thermocatalyst has already be proposed and used for carbon dioxide hydrogenation (Tuning of Catalytic Activity by Thermoelectric Materials for Carbon Dioxide Hydrogenation, *Advanced Energy Materials*, 2018, 8, 1701430; Thermoelectric avenue by James Gallagher). Having said that, the paper present interest to the reader for ROS, but not enough for the "Thermoelectric avenue". In the revision, the so-tightly related refs should be added. Below are my comments and questions.

Re: We would like to thank the reviewer for the detailed comments and suggestions regarding our manuscript. We have revised the manuscript following the guidelines mentioned by the reviewer and the point-by-point responses are listed below. We also thank the reviewer for mentioning some valuable references which have been added in the revised manuscript.

Added references:

28. Achour A, Chen K, Reece MJ, Huang Z. Tuning of catalytic activity by thermoelectric materials for carbon dioxide hydrogenation. *Advanced Energy Materials* **8**, 1701430 (2018).
29. Gallagher J. Thermoelectric Avenue. *Nature Energy* **2**, 834-834 (2017).

What is the mean of “w/o cooling and w/ cooling” in Figure 4. The author demonstrated a practical application for the thermocatalyst in Figure 5. From the text, it is not clear that how did they obtain the E.coli of 40%? Did you do the experiment via cooling fan or a hair dryer, as that in Figure 4?

Re: Thanks to the reviewer for his valuable comments and suggestions. Authors wish to clarify that for the disinfection performance, experiments were conducted on bare CFF and Bi₂Te₃@CFF. In order to create a temperature difference across the antibacterial filter to trigger the thermocatalytic activity, the bare CFF and Bi₂Te₃@CFF were exposed to cold air in one set of experiment and in another set of experiment, no cold air was provided (just room temperature). Obtained results indicate that without cooling almost no disinfection performance was observed while with cooling the disinfection performance was significant. So, “w/o cooling” and w/cooling” represent the experimental conditions having no temperature difference and a cooling fan induced temperature difference, respectively. For better understanding of the fact, the concerned issue has been modified in Figure 5 of the revised manuscript.

Moreover, authors apologize to the reviewer for not being able to clarify the bacteria survival percentage in the practical application as shown in Figure 5 of original manuscript. During the

experiment, initially, a bacterial suspension (2×10^6 CFU/mL) was added to Bi₂Te₃@CFFs and then it was treated under the temperature difference created by the air conditioner. The temperature of air-conditioner was set to 17 °C so that the applied temperature difference is similar to that of cooling fan experiment. Control experiments were also performed under the same conditions without the temperature difference. Both the treated and untreated Bi₂Te₃@CFFs were immersed in 0.85% sodium chloride solution from which 100 μL of aliquots were collected from each group and plated on aseptic plates. The bacterial colonies were counted from the plates after 24 hrs of incubation at 37 °C. The survival rates were determined by using the formula $C/C_0 \times 100 \%$ where C_0 is the concentration of the bacteria solution before thermal treatment and C is the concentration of the remaining bacteria after the thermal treatment. Compared to the non-treated filter, the *E. Coli* survival rate was obtained to be 40 % which means 60 % of the bacteria were killed by the thermocatalytic activity in the air conditioner. This discussion is now added in the revised manuscript.

Changes in the manuscript:

This test was conducted by using an air conditioner. Bi₂Te₃@CFF with dimensions of 8×15 cm was mounted on an indoor part of a split air conditioner. The temperature of the air conditioner was set to 17 °C in order ensure similar range of temperature difference as in case of cooling fan experiment. In this case the same bacterial concentration (2×10^6 CFU/mL) was used to react the Bi₂Te₃@CFF as in the case of cooling fan experiment.

Disinfection test by antibacterial filter

To demonstrate the performance of Bi₂Te₃@CFF as an antibacterial filter, the disinfection test was conducted using a hair dryer and a cooling fan. At first 1 mL of bacterial solution (2×10^6 CFU/mL) was added to 1×1 cm Bi₂Te₃@CFFs and treated for 20 mins under the temperature difference crated by hair dryer and cooling fan. Control experiments were also performed under the same conditions without the temperature difference. Both the treated and untreated Bi₂Te₃@CFFs were immersed in 1 mL of 0.85% sodium chloride solution. Aliquots of 100 μL of bacterial solution were collected from each group and plated on an aseptic plates. The bacterial colonies were counted from the plates after 24 hrs of incubation at 37 °C. The survival rates were determined by using the formula $C/C_0 \times 100 \%$ where C_0 is the concentration of the bacteria solution before thermal treatment and C is the remaining concentration of the bacteria after the thermal treatment.

If yes for the above question (seems yes, because they said the treatment was the same as the air for the disinfection test the), what is the real temperature gradient for the antibacterial filter in the air conditioner, and what is the E.coli values? A real demo is necessary for this concept, because the demo in Figure 4 with a huge temperature gradient of >10 K.

Re: Authors apologize that the real temperature gradient for the air conditioner experiment is difficult to measure. However, the results obtained in the cooling fan experiment can be compared to the air conditioner experiment as both the experiments were carried out in similar environment. Also, the E.coli concentration for air conditioner experiment was 2×10^6 CFU/mL which was identical to the bacterial concentration used for cooling fan experiment.

Authors also wish to clarify that for the cooling fan experiment as shown in figure 4 (original manuscript), the temperature difference was around 8 K (front side at 26 °C and back side at 34 °C). Since the air conditioner was set at 17 °C, it can be concluded that almost similar range of temperature difference was achieved when the disinfection tests were performed at air conditioner. The fact is also clear from the antibacterial activity exhibited by Bi₂Te₃@CFF which clearly shows that significant amount of bacterial disinfection is realized under air conditioner. However, for the hot wind test we agree that the temperature difference is a little large (>10 K) but with a larger temperature difference, the disinfection performance has also increased in hot wind experiment. The related discussion is now added in the revised manuscript.

Changes in the manuscript:

The temperature inside the air conditioner was lower than the air in the surroundings, creating a temperature gradient to trigger the thermocatalytic reaction. Since the air conditioner was set at 17 °C, it can be concluded that the temperature difference across the Bi₂Te₃@CFF was similar to that in the case of cooling fan experiment. This fact is also clear from the disinfection results which reveal that thermocatalytic reaction at air conditioner also leads to ~ 60% bacterial degradation as in the case of cooling fan experiment.

It is also interesting to note that since the achieved temperature difference across the Bi₂Te₃@CFF is higher for hot air treatment, the bacterial survivability (~ 28%) is also less in this condition as compared to cooling fan experiment.

In Figure 5, the two SEM images are presented in order to reveal the reusability of the thermocatalyst. However, it might be not from the same zone, and why the inset of the SEM images are with different size scale, 2 um and 4 um??? The SEM images are absolutely not enough to show the reusability of thermocatalyst, XRD and other characterization are necessary.

Re: Thanks to the reviewer for his valuable comments and suggestions. As per reviewer's suggestion the SEM images of same scale size is provided in the revised manuscript. Moreover, as per reviewer's recommendation, the XRD and Raman spectra of the Bi₂Te₃@CFF before and after 30 days of operation under air conditioner are also included in the Figure S14 of the revised manuscript. Even after the 30 days of catalytic reaction, the Bi₂Te₃@CFF retains their initial spectral characteristics which readily proves the long term structural stability of the antibacterial filter. The concerned sections are included in the revised manuscript.

Changes in the manuscript:

XRD and Raman spectra of the $\text{Bi}_2\text{Te}_3@\text{CFF}$ (Figure S14) also indicate that even after 30 days of thermocatalytic operation under air conditioner, the antibacterial filter retains their initial spectral characteristics.

Line 193, "...revealing that the amount of H_2O_2 generated by Bi_2Te_3 NPs was at least 20 times higher than the amount generated by bulk Bi_2Te_3 ." More discussion are necessary to the readers.

Re: Authors are thankful to the reviewer for his valuable suggestions. It is well-known that nanomaterials possess much higher surface area than their bulk counterparts and this higher surface area is beneficial for boosting the catalytic reactions (*please see reference 43, 44 of revised manuscript*). In the present research, the 2D plate like structure of Bi_2Te_3 NPs facilitates high surface area and consequent 20 times higher thermocatalytic performance than bulk Bi_2Te_3 . The related discussion has been added in the revised manuscript.

Changes in the manuscript:

Since catalysis is a surface phenomenon, high surface area of the catalysts can effectively boost the surface chemical reactions by increasing the active catalytic sites. The 2D plate like structure of Bi_2Te_3 NPs facilitates higher surface area than bulk Bi_2Te_3 which is responsible for this superior thermocatalytic activity. The nanostructuring approach for realizing high surface area and efficient catalytic activity are already widely reported in the literature ^[43, 44].

In Figure 1, from the thermocatalysis mechanism for H_2O_2 generation, the generation of $\cdot\text{O}_2^-$ is necessary. The authors should present the evidence for the mechanism. In my opinion, the concentration of $\cdot\text{O}_2^-$ by different temperature gradient or materials should be provided.

Re: Thanks to the reviewer for his critical comments. As per reviewer's suggestion, generation of superoxide radicals ($\cdot\text{O}_2^-$) for different temperature differences was probed through XTT (2, 3-bis (2- methoxy-4-nitro-5-sulfophehyl)-2H-tetrazolium-5-carboxanilide) assay and the results are shown in Figure S10 of the revised manuscript. Figure S10 clearly indicates that as the temperature difference increases, Bi_2Te_3 NPs generate more and more superoxide radicals. Similar trend is also visible from the superoxide radical generation by bulk Bi_2Te_3 as shown in Figure S3. The related discussion is included in the revised manuscript.

Changes in the manuscript:

The generation of superoxide radicals during the thermocatalysis reaction was estimated quantitatively by using XTT (2, 3-bis (2- methoxy-4-nitro-5-sulfophehyl)-2H-tetrazolium-5-carboxanilide) assay. Typically, aqueous dispersion of bulk and Bi_2Te_3 NPs was mixed with XTT (50 μM) and then, the solution was subjected to different temperature gradient at water bath. After the reaction Bi_2Te_3 was separated by centrifugation and the absorbance spectra of supernatants were monitored at 470 nm.

It is also found that the increase in temperature difference boosts the thermocatalytic generation of superoxide radicals which is evidenced from Figure S10.

How about the bio-capability of Bi_2Te_3 NPs? Is that health for air condition filter?

Re: Authors are thankful to the reviewer for his meaningful comment. As per author's suggestion, both *in-vitro* and *in-vivo* biocompatibility of the Bi_2Te_3 NPs was investigated by collaborating with Dr. Fu-Cheng Kao (Chang Gung Memorial Hospital, Taiwan). The *in-vitro* cytotoxicity analysis of the Bi_2Te_3 NPs was performed using the Alamar blue assay. Here, the proliferation of the HaCaT cells was investigated by comparing the viability of the cells in presence and absence of

Bi_2Te_3 @CFFs. After 24 hrs of incubation, it was found that the cells in presence of Bi_2Te_3 @CFFs maintained ~100 % viability even at the high concentrations, and there is a negligible difference compared to the control group. In addition, the *in-vivo* biocompatibility test performed on the rat wound model clearly demonstrates that there are no signs of bacterial infection or tissue damage observed even after 3 days of incubating the wound area with Bi_2Te_3 @CFFs at concentration of 1500 $\mu\text{g/ml}$. These results clearly indicate that the as-prepared Bi_2Te_3 NPs are highly biocompatible and are safe for usage as a filter for the air conditioner.

The authors would also like to emphasize that the focus of this manuscript is solely for environmental applications which would require comparatively lesser biocompatibility suggesting that the Bi_2Te_3 NPs is extremely healthy for air filter applications.

Authors wish to mention that the animal use protocol has been reviewed and approved by the Institutional Animal Care and Use Committee (IACUC), and the committee recognises that the proposed animal experiment follows *the Animal Protection Law by the Council of Agriculture Executive Yuan, R.O.C* and the guidelines as shown in the Guide for the Care and Use of Laboratory Animals as promulgated by the Institute of Laboratory Animal Resources, National Research Council, U.S.A. Authors also wish to mention that the cell lines were obtained from American Type Culture Collection (ATCC, Manassas, VA, USA).

In the title, "Highly efficient" was used, related discussion should be addressed.

Re: Authors are thankful to the reviewer for his important comment. Taking cue from reviewer's suggestion, the title of the manuscript has been changed and term "Highly efficient" has been removed. The modified title of the manuscript is "Thermocatalytic Hydrogen Peroxide Generation by Bi_2Te_3 Nanoplates for Bacterial Disinfection Applications".

Reviewer #2 (Remarks to the Author):

In this work, the authors reported Bi₂Te₃ thermocatalyst for hydrogen peroxide generation and bacterial disinfection applications. Though there are some points need to be clarified, it is an interesting work that may finds possible application in the future.

Re: Authors would like to thank the reviewer for his valuable comments and kind consideration. We have revised the manuscript following the suggestions mentioned by the reviewer and the point-by-point responses are listed below.

1. In line 126-127, "...reaction is strongly correlated with the figure of merit (zT value) and the dimension of the thermoelectric material. The zT value of PbTe is much smaller than both Bi₂Te₃ and Sb₂Te₃ in the low temperature range." Since zT value is a very important factor to consider for thermocatalyst, the authors provide the zT values of samples (coated on CFF) used in this study.

Re: Thanks to the reviewer for his valuable comments. Authors fully agree with the reviewer that for thermoelectric materials zT value is a very important factor. However, particularly for thermocatalysis applications, thermoelectric voltage and Seebeck coefficient of the catalyst is more important which is also reflected from the next comment of the reviewer (*please see reference 37, revised manuscript*). In fact, the generated thermoelectric voltage facilitates efficient carrier separation within the thermocatalyst which plays a pivotal role in achieving proficient thermocatalytic performance. Moreover, authors also wish to emphasize that in the current research, CFF acts only as filter and the obtained thermocatalytic performance is achieved due to the thermoelectric properties possessed by Bi₂Te₃ NPs. Hence, taking cue from the reviewer's comment, authors have measured the thermoelectric voltage and Seebeck coefficient of Bi₂Te₃ NPs and the obtained results are appended below. Synthesized Bi₂Te₃ NPs demonstrate a Seebeck coefficient of 497 $\mu\text{V/K}$ which is comparable with the Seebeck coefficient values already reported in literature.

2. Following the above comment, the underlying mechanism of thermocatalyst is not very clear stated in the present work. The authors use Fig.1a to explain the mechanism. Based on this model, what is important for thermocatalysis should be the thermoelectric voltage. Therefore, the authors should compare the Seebeck coefficient of their materials since the thermoelectric voltage generated on two ends of their samples is proportional to the Seebeck coefficient, while zT is related to the power output. Besides, the authors mistakenly treat temperature difference as temperature gradient. What the authors measured are actually temperature difference, not gradient and it would be very difficult to accurately measure the gradient in the current work if not impossible. Fortunately, the thermoelectric voltage depends on the temperature difference, not the gradient.

Re: Authors are thankful to the reviewer for his valuable comments and suggestions. Authors are in complete agreement with the reviewer that for thermocatalytic applications thermoelectric voltage is an important parameter in determining the catalytic activity. Following reviewer's suggestion authors have measured the generated thermoelectric voltage and Seebeck coefficient

of the synthesized Bi₂Te₃ NPs. The calculated value of Seebeck coefficient for Bi₂Te₃ NPs is 497 $\mu\text{V}/\text{K}$. Literature review also reveals that reduction in materials dimension is already reported as an effective approach for enhancing the thermoelectric properties of the materials (*Physical Review Letters* **94**, 096601 (2005), *J. Appl. Phys.* **128**, 035108 (2020)). On the other hand, the Seebeck coefficient of bulk thermoelectric materials such as Bi₂Te₃, PbTe and Sb₂Te₃ are widely reported in the literature which clearly indicates that the Seebeck coefficient of PbTe and Sb₂Te₃ are much less than Bi₂Te₃ (reference 30-32, revised manuscript). The obtained Seebeck coefficient readily validates the obtained results that H₂O₂ generation is highest for Bi₂Te₃ as compared to other thermoelectric materials (PbTe and Sb₂Te₃) used in this research. Moreover, authors also agree with the reviewer that in actual experiment the temperature difference was measured rather than the temp gradient. Following reviewer’s suggestion, the concerned sections have been modified in the revised manuscript.

Changes in the manuscript:

Furthermore, the amount of H₂O₂ generated in the thermocatalytic reaction is strongly correlated with the Seebeck coefficient^[37] and the dimension of the thermoelectric material. The Seebeck coefficient of both Sb₂Te₃ and PbTe is much smaller than Bi₂Te₃ in the low temperature range^{[30-}

^{32]}, resulting in the highest amount of H₂O₂ production in case of Bi₂Te₃. As any catalytic reaction is accelerated with a large surface area, small size of Bi₂Te₃ are also responsible for demonstrating superior thermocatalytic effect than Sb₂Te₃ and PbTe (Figure S1).

Since the thermoelectric voltage under a particular temperature difference plays the key role in influencing the thermoelectric effect, a thermoelectric material whose Seebeck coefficient is high should be selected for achieving efficient thermocatalytic performance. Near room temperature, bismuth telluride (Bi₂Te₃) has a higher Seebeck coefficient ^[30-32] than other common thermoelectric materials such as antimony telluride (Sb₂Te₃) and lead telluride (PbTe) due to its outstanding physical properties and distinctive electronic structure ^[33].

Since the Seebeck coefficient of a thermoelectric material plays the pivotal role in controlling the thermocatalytic activity, the thermoelectric voltage and corresponding Seebeck coefficient of Bi₂Te₃ NPs are also estimated (Figure 3). The thermoelectric device fabricated by Bi₂Te₃ NPs was subjected to a series of temperature difference by adjusting the temperature (40 to 100 °C) of a hot plate. When the temperature of hot end of the device is increased from 40 to 100 °C, the generated thermoelectric voltage is also enhanced from 0.72 mV to 19.83 mV (Figure 3a & b). The actual temperature difference created across the device is represented by the temperature distribution images as shown in Figure 3a. The developed thermoelectric voltage is a consequence of temperature difference induced carrier migration within the Bi₂Te₃ NPs. The obtained thermoelectric voltages (ΔV) are plotted as a function of actual temperature difference (ΔT) and the slope obtained from the linear fitting of ΔV vs. ΔT curve denotes the Seebeck coefficient (S) (Figure 3c). The synthesized Bi₂Te₃ NPs exhibit a high Seebeck coefficient of $\sim 497 \mu\text{V/K}$ which strongly validates the potential of Bi₂Te₃ NPs to act as efficient thermocatalyst. It is noteworthy that when one end of the thermoelectric device is cooled keeping the other end at room temperature, the charge carriers within Bi₂Te₃ NPs move to a reverse direction and consequently the opposite output voltage signals are obtained (Figure S7). Moreover, it is also important to note that obtained thermoelectric voltage for applied +10 K temperature difference is higher (0.45 mV) than the generated thermoelectric voltage (0.19 mV) for -10 K temperature difference (Figure S7a). The impact this phenomenon is vividly reflected from the H₂O₂ generation performance of Bi₂Te₃ NPs at different temperature differences as shown in Figure 2d. The amount of H₂O₂ generated under a -10 K temperature difference (8.5 μM) is lower than that generated under a 10 K temperature difference (18 μM). Obtained results strongly signify that generated thermoelectric voltage of a thermocatalyst plays the crucial role in controlling surface electrochemical reaction and consequent catalytic activity.

Measurement of Seebeck coefficient:

A device size of 1cm² was used for the thermoelectric voltage measurement and a hot plate was used to create the temperature difference. For measuring the thermometric voltage below room temperature, a PID controlled cooling stage was used. The temperature difference between the surface of the hot-plate/cooling stage and the top surface of device induces temperature difference

across the device and the consequent thermoelectric voltage is generated. The generated thermoelectric voltages were recorded with a low noise voltage preamplifier (SR-560). To determine the actual temperature difference created across the device, the temperature at the surface of the hot-plate (T1) and top of the device (T2) were monitored with the help of IR camera.

3. To assist the model mechanism proposed in Fig.1a, detailed experiments should be conducted to determine the band edge positions of samples to verify that the H_2O_2 generation reaction is at least thermodynamically possible.

Re: Thanks to the reviewer for his meaningful comments. As per reviewer's suggestion, we have determined the band edges of Bi_2Te_3 by employing high resolution valence band XPS which shows that the valence band maxima is located at -0.41 eV. The binding energy position corresponding to zero electronic density of state represents the valence band potential in the XPS valence band spectra. The obtained valence band potential is in close agreement with the value already reported in the literature (*reference 39, revised manuscript*). Conduction band position are calculated from the equation $E_{\text{CB}} = E_{\text{VB}} - E_{\text{g}}$ (E_{g} , E_{CB} and E_{VB} indicate the band gap energy, conduction band potential and valence band potential). Considering the band gap of Bi_2Te_3 as 0.2 eV, the conduction band potential is calculated to be -0.61 eV. The obtained conduction band potential is more negative than the redox potential of $\text{O}_2/\cdot\text{O}_2^-$ (-0.33 eV vs. NHE) (*reference 41, 42, revised manuscript*). The more negative conduction band potential readily implies that formation $\cdot\text{O}_2^-$ radical and consequent H_2O_2 generation is thermodynamically possible. The related discussion is now added in the revised manuscript.

Changes in the manuscript:

Moreover, in order to prove that generation of superoxide radical is thermodynamically possible, the band edge position of Bi_2Te_3 NPs was determined with the help of high resolution valence

band XPS (Figure S11). The valence band maxima (VBM) is determined from the binding energy position at which the electronic density of states become zero. From Figure S11 it is clear that the VBM position of Bi₂Te₃ NPs is located at -0.41 eV which is in close agreement with the VBM position of Bi₂Te₃ reported previously [39]. Conduction band minima (CBM) position are calculated from the equation $E_{CB} = E_{VB} - E_g$ (E_g , E_{CB} and E_{VB} indicate the band gap energy, conduction band potential and valence band potential). Considering the band gap of Bi₂Te₃ as 0.2 eV [40], the CBM position is calculated to be -0.61 eV. The obtained conduction band potential is more negative than the redox potential of O₂/·O₂⁻ (-0.33 eV vs. NHE) [41, 42]. The more negative conduction band potential readily implies that formation ·O₂⁻ radical and consequent H₂O₂ generation is thermodynamically possible.

4. The thermoelectric materials are narrow bandgap semiconductors. Are they also photocatalysts for H₂O₂ generation? If yes, how to avoid the interference of photocatalysis during the experiment?

Re: Thanks to the reviewer for his valuable comments. In order to eliminate any possible influence from photocatalysis, in the current work the thermocatalytic reactions were conducted in dark. Moreover, to investigate the effect of photocatalysis on Bi₂Te₃, the thermocatalysis experiments were also conducted in presence of light. Obtained results not only indicate that ambient light has negligible effect in the thermocatalytic activity offered by Bi₂Te₃ but also it supports the practical applicability of the thermocatalysts in the real circumstances. Particularly, for the thermocatalysts, temperature difference induced thermoelectric voltage generation and consequent band bending is primarily responsible for reducing the energy difference between band energy and redox potential which is eventually not possible for a photocatalyst. Hence the obtained thermocatalytic performance remains almost unaffected from the interference of photocatalysis. The related discussion has been included in the revised manuscript.

Changes in the manuscript:

Moreover, since Bi_2Te_3 is a narrow band gap semiconductor, it is necessary to find out if there is any contribution from photocatalysis in the thermocatalytic activities shown by Bi_2Te_3 NPs. In presence of ambient light, Bi_2Te_3 NPs exhibit almost similar catalytic performance as compared to dark (Figure S12). Actually, for the thermocatalysts, temperature difference induced thermoelectric voltage generation and consequent band bending is primarily responsible for reducing the energy difference between band energy and redox potential which is eventually not possible for a photocatalyst. Hence the obtained thermocatalytic performance remains almost unaffected from the interference of photocatalysis. These phenomena not only indicate that ambient light has negligible effect in the thermocatalytic activity offered by Bi_2Te_3 but also support the practical applicability of the thermocatalysts in the real circumstances.

5. In line 136, “As the number of thermal cycles (C1, C2, and C3) increases...” It is better to provide accurate cycle number such as 20, or 30. Besides, there is no experimental details to tell how the bulk materials (Bi_2Te_3 and others) are prepared and tested in Fig.1b-1d.

Re: Thanks to the reviewer for his valuable suggestions. Authors wish to clarify that, in the current work thermocatalytic performance was investigated for 3 thermal cycles. Hence, C1, C2 and C3 represents 1st, 2nd and 3rd cycle of catalytic experiment. Moreover, authors also wish to mention that the bulk thermoelectric materials (Bi_2Te_3 , PbTe and Sb_2Te_3) of purity > 99% were directly purchased from Alfa Aesar. The concerned sections are modified in the revised manuscript.

Changes in the manuscript:

Therefore, as an attempt to verify this concept, three kinds of commercially available bulk thermoelectric materials, Bi_2Te_3 , Sb_2Te_3 and PbTe (purchased from Alfa Aesar), were used for the thermocatalytic reactions (Figure 1b).

As the number of thermal cycles increases from 1 to 3, the disinfection performance of bulk Bi_2Te_3 increases gradually.

Thermocatalysis experiments were conducted in a water bath where the thermocatalysts in the contaminant solutions were subjected to 3 thermal cycles denoted as C1, C2 and C3.

6. In the experiment section, according the statement “In each cycle, the materials at first were allowed to react at specific temperature (15/35/45) for 5 min and then were returned to room temperature (RT) for 5 min...” It seems that the temperature gradient was obtained by immerse the sample in hot and cold water alternatively. I wonder the temperature difference across the nanoplate would be large enough to have some real effect in catalysis, considering the thickness of Bi_2Te_3 nanoplate is in the nano-range. Besides, the temperature difference will soon diminish due to thermal conduction. It would be more convincing if a constant temperature difference across thermoelectric sample can be maintained.

Re: Authors wish to clarify that, the disinfection tests were performed by keeping the contaminant solution containing the thermocatalyst at first at a particular temperature (15/35/45 °C) in water bath for 5 min and after 5 min, it was kept at room temperature (25 °C) for another 5 min.

Authors also completely agree with the reviewer that for nanomaterials it is difficult to maintain the temperature difference and even if the temperature difference is created once, thermal conduction strongly contributes in neutralizing that. In order to overcome this barrier, in the current work we used cyclic heating operation where the thermocatalyst restores their initial temperature when they are kept at room temperature after each heating step. In order to establish this, we have carried out control experiments where the contaminant solution with the thermocatalyst were allowed to react under a particular temperature (35 °C) for 15 min at a stretch. As the reaction time increases, the rate of increase in H₂O₂ generation becomes almost negligible as compared to cyclic heating condition (Figure S8, revised manuscript). This phenomena strongly indicate that cyclic heating condition is more effective for thermocatalytic ROS generation than continuous heating operation. Hence, for further experiments cyclic heating conditions were used.

Changes in the manuscript:

It is also interesting to note that H₂O₂ generation efficiency significantly higher in case of cyclic heating condition than the continuous heating condition (Figure S8). Since, for nanomaterials it is difficult to maintain the temperature difference for a long time in the current work cyclic heating operation was used in which the thermocatalyst restores their initial temperature when they are kept back at room temperature after each heating step.

7. In the disinfection test, the temperature on the back side of the filter in Fig.4b and 4c is supposed to be the ambient temperature, which should be the same. Why is there a large difference in the ambient temperature, which is ~ 34 degree in Fig.4b and ~28 degree in Fig.4c? How long was the disinfection test conducted?

Re: Thanks to the reviewer for his critical comments. It is to be noted that the two types of disinfection tests (with cold air and hot air) were carried out on two different places where there was a variation in the room temperature. For this reason, a difference in room temperature is noticed in Figure 4b and c of the original manuscript. Authors also wish to mention that total 20 mins of treatment time was selected for both the disinfection tests under hot air and cold air.

Changes in the manuscript:

At first 1 mL of bacterial solution (2×10^6 CFU/mL) was added to 1×1 cm Bi_2Te_3 @CFF and further treated for a total time of 20 mins under the temperature difference created by hair dryer and cooling fan.

8. For air conditioner antibacterial filter experiment. It is better to provide the result of a control experiment using a non-coated CFF filter for comparison. Control experiment using bare CFF sample should also be conducted in Fig.2d for comparison.

Re: Thanks to the reviewer for his valuable suggestions. Taking cue from reviewer's comment, thermocatalytic control experiment was performed using bare CFF under different temperature difference which is shown in Figure S13 of the revised manuscript. From the figure it is clear that bare CFF can not produce H_2O_2 under any applied temperature difference which indicates that bare CFF has no antibacterial activity. Moreover, authors also wish to emphasize that disinfection test under both cold air (Figure 5d, revised manuscript) and air-conditioner (Figure 6c, revised manuscript) were performed by maintaining similar range of temperature difference ($\sim 8\text{K}$). Since the disinfection test under cold air exhibits that bare CFF has negligible effect in antibacterial activity, it can be readily concluded that it will not show any disinfection performance under air-conditioner. In order to keep the discussion streamline, the thermocatalytic H_2O_2 generation performance by bare CFF is demonstrated in Figure S13 instead of Figure 2d. The concerned discussion is also included in the revised manuscript.

Changes in the manuscript:

However, bare CFF does not exhibit any H_2O_2 generation under the applied temperature difference.

The commercial unmodified CFF does not exhibit any antibacterial activity even upon the application of a temperature difference which is fully consistent with the poor H_2O_2 generation performance shown by bare CFF as represented in Figure S13.

9. It is meaningless to show the total H_2O_2 generation in the experiments in Fig.1 and 2. Instead, the H_2O_2 generation per unit mass of thermoelectric material should be shown.

Re: Authors are thankful to the reviewer for his important suggestion. As per reviewer's suggestion the H_2O_2 generation efficiency is included in Figure 1 and 2 of the revised manuscript.

10. Lines 110-112, when the valence and conduction bands are tilted, how can both of them come close to the redox potential for generating superoxide?

Re: Thanks to the reviewer for his valuable comment. Authors wish to clarify that the temperature gradient induced thermoelectric voltage generation leads to the bending of both the valence band and conduction band in the thermocatalysts. However, since the position of only conduction band is energetically favorable for producing $\cdot\text{O}_2^-$ from O_2 , the energy difference between the conduction band and redox potential of $\text{O}_2/\cdot\text{O}_2^-$ decreases significantly. As a result, electrons from the thermocatalyst can easily transfer to the solution to produce ROS. The related sections have been modified in the revised manuscript.

Changes in the manuscript:

To follow this variation in energy, both the valence band and conduction band tilt across the material and the conduction band comes very close to the redox potential for generating superoxide ($\cdot\text{O}_2^-$) radicals.

Finally, the authors are grateful to the reviewers for their meaningful comments which helped to improve the overall quality of the manuscript.

Reviewers' Comments:

Reviewer #1:

Remarks to the Author:

The revision is satisfied, and all the concerns have been addressed.

Reviewer #2:

Remarks to the Author:

The authors have addressed most of my questions satisfactorily. However, I still have one more question which needs to be clarified. The production of ROS is related to the charge transfer from the conduction band electrons. This is a half-reaction. There must be another half-reaction taking place to fill electrons into the hole in the valence band so that the whole reaction can be sustained. What is the other half-reaction? Did you use hole-scavengers or sacrificial agents in the production of ROS?

Reply to the reviewers' comments

Reviewer #1 (Remarks to the Author):

The revision is satisfied, and all the concerns have been addressed.

Re: We would like to thank the reviewer for his positive evaluation and further considering our manuscript for publication.

Reviewer #2 (Remarks to the Author):

The authors have addressed most of my questions satisfactorily. However, I still have one more question which needs to be clarified. The production of ROS is related to the charge transfer from the conduction band electrons. This is a half- reaction. There must be another half-reaction taking place to fill electrons into the hole in the valence band so that the whole reaction can be sustained. What is the other half-reaction? Did you use hole-scavengers or sacrificial agents in the production of ROS?

Re: Thanks to the reviewer for his valuable comments. Authors agree with the reviewer that holes are created in the valence band once electrons are transferred to the conduction band of Bi_2Te_3 thermocatalyst. In one half of the reaction, the generated electrons in conduction band react with the dissolved O_2 to produce superoxide radicals ($\cdot\text{O}_2^-$). In ideal case, the generated holes in the valence band reacts with OH^- ions to produce hydroxyl radicals ($\cdot\text{OH}$) which constitutes the other half reaction. However, the position of valence band maxima of Bi_2Te_3 is located at -0.41 eV (Supplementary Fig. 11) which is not sufficiently positive to oxidize OH^- to $\cdot\text{OH}$ as the redox potential of $\text{OH}^-/\cdot\text{OH}$ is located at 1.99 eV vs. NHE (*Advanced Energy Materials*. 2017, 7, 1700025). Hence, because of this unfavorable band position, holes cannot take part in the thermocatalytic reaction. This type of catalytic reaction where one half of the reaction is not feasible, is widely reported in the literature and is considered as the prime challenge for realizing highly efficient catalyst (*Angew. Chem. Int. Ed.* 2013, 52, 7372-7408, *Nature Communications* (2020) 11, 3043). In this study we have not used any hole-scavengers or sacrificial agents, however, sacrificial donors can be used in future to increase the thermocatalytic efficiency of the Bi_2Te_3 thermocatalysts.

Finally, authors are thankful to the reviewers for their critical comments which helped to improve the overall quality of the manuscript.